# The Antivirulence Activity of Umbelliferone and Its Protective Effect against *A. hydrophila*-Infected Grass Carp

**DOI:** 10.3390/ijms231911119

**Published:** 2022-09-22

**Authors:** Ling Zhao, Xiaoyu Jin, Ziqian Xiong, Huaqiao Tang, Hongrui Guo, Gang Ye, Defang Chen, Shiyong Yang, Zhongqiong Yin, Hualin Fu, Yuanfeng Zou, Xu Song, Cheng Lv, Wei Zhang, Yinglun Li, Xun Wang

**Affiliations:** 1College of Veterinary Medicine, Sichuan Agricultural University, Chengdu 611130, China; 2Institute of Animal Genetics and Breeding, College of Animal Science and Technology, Sichuan Agricultural University, Chengdu 611130, China

**Keywords:** umbelliferone, *A. hydrophila*, grass carp, antivirulence, transcriptome

## Abstract

*A. hydrophila* is an important pathogen that mainly harms aquatic animals and has exhibited resistance to a variety of antibiotics. Here, to seek an effective alternative for antibiotics, the effects of umbelliferone (UM) at sub-MICs on *A. hydrophila* virulence factors and the quorum-sensing system were studied. Subsequently, RNA sequencing was employed to explore the potential mechanisms for the antivirulence activity of umbelliferone. Meanwhile, the protective effect of umbelliferone on grass carp infected with *A. hydrophila* was studied in vivo. Our results indicated that umbelliferone could significantly inhibit *A. hydrophila* virulence such as hemolysis, biofilm formation, swimming and swarming motility, and their quorum-sensing signals AHL and AI-2. Transcriptomic analysis showed that umbelliferone downregulated expression levels of genes related to exotoxin, the secretory system (T2SS and T6SS), iron uptake, etc. Animal studies demonstrated that umbelliferone could significantly improve the survival of grass carps infected with *A. hydrophila*, reduce the bacterial load in the various tissues, and ameliorate cardiac, splenic, and hepatopancreas injury. Collectively, umbelliferone can reduce the pathogenicity of *A. hydrophila* and is a potential drug for treating *A. hydrophila* infection.

## 1. Introduction

*A. hydrophila* is a common opportunistic fish bacterial pathogen, one of the main severe threats to aquaculture, and is prevalent worldwide. This disease exhibits a high fatality rate and can cause septicemia in carp [1], perch [2], catfish [3], and grass carp; red-leg syndrome in frog; and red sore disease in perch and carp [4]. There are many factors affecting the pathogenicity of *A. hydrophila*, including surface polysaccharide, surface layer, flagella, iron-binding system, exotoxin and extracellular enzyme, secretion system, biofilm, and motility [5]. When environmental conditions are favorable, planktonic *A. hydrophila* frequently adapts to their surrounding environment and causes disease by secreting various virulence factors. When environmental conditions deteriorate, the biofilm produced by *A. hydrophila* can provide a protective effect on autochthonous bacteria. Bacteria residing in the biofilm can continuously secrete proteolytic enzymes and adhesins to enhance bacterial virulence and antibiotic resistance [6].

Over the past many years, the strategy for treating and preventing *A. hydrophila* infection mainly relied on antibiotics. Long-term antibiotic use had generated pan-resistance to sulfonamides, chloramphenicols, tetracyclines, streptomycin, ampicillin, and other antimicrobials. *A. hydrophila* strains isolated from zebrafish in Seoul, Korea, were found to be resistant to amoxicillin (100%), naphthalic acid (100%), ampicillin (93.02%), tetracycline (74.42%) and imipenem (65.15%) [7], and the strains isolated from goldfish in India were also resistant to multiple antibiotics [8]. Additionally, the abuse of antibiotics can damage water bodies and the structure of intestinal flora in aquatic animals, further increasing the difficulty of aquatic disease prevention and treatment.

Plant-derived bioactive constituents with vigorous antivirulence activity have become an essential source of potential drugs for treating bacterial infections. Early studies have shown that coumarins, critical secondary metabolites in various plants, are potential antibacterial agents. Several pyrenylated coumarins and furanocoumarins extracted from the roots of *Prangos hulusii* exhibit activity of bacterial growth suppression [9]. Daphnetin (7,8-dihydroxycoumarin) has been demonstrated to exert anti-*H.pylori* activity by preventing *H. pylori* colonization of the stomach [10]. Benzothiazole–1,2,3- triazole–coumarin showed good anti-*Moraxella catarrhalis* potency (MIC ≤ 0.25 μg/mL) comparable to that of reference antibiotic azithromycin [11]. Coumarins mainly attenuate the pathogenicity of pathogens by inhibiting bacterial virulence factors, decreasing biofilm formation, or regulating quorum-sensing (QS) signals. Gutierrez-Barranquero et al. reported that coumarin could inhibit biofilm formation in *P. aeruginosa*, *E. coli*, and *Edwardii*, protease activity in *S. maltophilia* and *B. cepacia*, and bioluminescence in *A. fischeri* [12]. Lee et al. suggested that the inhibition of coumarin and umbelliferone on biofilms might be related to the reduction of QS-controlled *lsrA* gene expression [13].

Umbelliferone(7-hydroxycoumarin) is a kind of coumarin derivative with a hydroxyl substitute at position C7 widely derived from plants. Among various coumarin derivatives, hydroxycoumarin is considered a type with relatively prominent antimicrobial activity. Yang Liang et al. tested the inhibitory effects of 18 kinds of coumarins against the plant pathogen *R. solanacearum* and found that umbelliferone had better antibacterial activity than coumarin [14]. Although there have been some studies on the activity of umbelliferone against bacteria such as *R. solanacearum* and *E. coli* [13], its effect on aquatic animal pathogens has not been reported. In our preliminary study, umbelliferone exhibited a significant inhibition effect on the quorum-sensing system of *A. hydrophila*, suggesting that umbelliferone has potential antivirulence activity. Here, we explored the effect of umbelliferone on the virulence factors of *A. hydrophila* and analyzed its potential mechanism through transcriptome information. Furthermore, the protective effect of umbelliferone on grass carp infected with *A. hydrophila* was studied. Our results demonstrated that umbelliferone had an obvious inhibitory effect on the pathogenicity of *A. hydrophila* in vitro and in vivo.

## 2. Results

### 2.1. Inhibition of Virulence of A. hydrophila by Umbelliferone

The MIC value of umbelliferone to *A. hydrophila* is 512 μg/mL. The growth curve (Figure 1B) shows that when the concentration of umbelliferone was lower than 128 μg/mL (1/4MIC), it had no significant effect on the growth of *A. hydrophila*. All subsequent experiments were performed at subinhibitory concentrations (16–128 μg/mL) of umbelliferone on *A. hydrophila*.

The effect of umbelliferone on *A. hydrophila* biofilm is shown in Figure 1C. Umbelliferone had an inhibitory effect on *A. hydrophila* biofilm formation after being treated for 12 h, 24 h, 48 h, and 72 h. At 48 h, the maximum effect of 53.84% on biofilm formation was observed at a concentration of 128 μg/mL. These results indicated that umbelliferone mainly interfered with biofilm maturation.

Figure 1D indicates the results of the hemolysis test. The *A. hydrophila* strain displayed remarkably strong hemolysis of rabbit erythrocytes. Umbelliferone could significantly reduce hemolysis induced by *A. hydrophila* (*p* < 0.001). When the umbelliferone concentration was 16, 32, 64, and 128 μg/mL, the inhibition rate was 10.92%, 30.65%, 77.23%, and 95.25%, respectively.

Swimming and swarming were regulated by the flagellar movement of bacteria and were considered closely related to biofilm formation. In the present study, we observed these types of motilities in the presence of the sub-MICs umbelliferone (Figure 1E,F). Swimming and swarming motility assays were performed on a semisolid medium containing 0.3% and 0.5% agar, respectively. Umbelliferone treatment inhibited swimming and swarming motility. After being incubated at 30 °C for 17 h, umbelliferone inhibited the swimming of bacteria in a dose-dependent manner, and the peak suppression was 59.52% at 128 μg/mL (1/4 MIC). After incubating at 30 °C for 24 h and 48 h, umbelliferone significantly inhibited the swarming at high sub-MICs, and the peak suppressions were 23% for 24 h and 47.3% for 48 h, respectively.

### 2.2. Effects of Umbelliferone on Quorum-Sensing Signals of A. hydrophila

Two experiments were conducted to observe the effects of umbelliferone on AHL and AI-2, which are two quorum-sensing signals of *A. hydrophila*. We investigated the influences of the coculture supernatant of *A. hydrophila* with umbelliferone on violacein production in the AHL indicator strain *C. violaceum* CV026 and bioluminescence intensity in the AI-2 indicator strain *V. harveyi* BB170. As shown in Figure 2A, compared with the control group, the color of violacein in the three drug treatment groups was significantly lighter, and almost no violacein could be observed in the high-concentration group (128 μg/mL). According to Figure 2B, compared with the control group, the bioluminescence intensity in *V. harveyi* BB170 was decreased significantly in the umbelliferone groups. These results indicated that umbelliferone could significantly inhibit the quorum-sensing system in *A. hydrophila*.

### 2.3. Transcriptome Changes in A. hydrophila after Umbelliferone Treatment

We obtained about 7.0 G of clean data by transcriptome sequencing. Approximately 93.80~95.58% of the clean data were mapped to the reference genome (Appendix A). Transcriptome analysis identified 1896 differentially expressed genes between umbelliferone treated and untreated *A. hydrophila*, of which upregulated and downregulated genes were 848 and 1048, respectively (Figure 3A, Appendix A). The results of hierarchical cluster analysis (Figure 3B) and principal component analysis (PCA) (Figure 3C) showed that six different samples were clustered into two groups according to DEGs expression data. The umbelliferone treatment groups were clustered into a subgroup and separated from the control groups. Intriguingly, treatment with umbelliferone significantly reduced the expression of genes related to bacterial exotoxin, the two-component system, and the secretion system (Table 1), while genes linked with ribosomes, the citric acid cycle, carbon metabolism, and oxidative phosphorylation were upregulated (Table 2). Functional enrichment analysis found that upregulated genes are mainly implicated in translation, biosynthetic-, and metabolic-related biological processes, and downregulated genes are involved in transport, signal transduction, biological regulation, etc. (Figure 3D, Appendix A).

### 2.4. qRT-PCR Validation of Transcriptome Data

To confirm the reliability of transcriptome sequencing, qRT-PCR was used to verify the expression of 12 randomly selected genes. All selected genes showed consistent results between RNA sequencing and qRT-PCR data (Figure 4).

### 2.5. Protective Effect of Umbelliferone on Grass Carp Infected with A. hydrophila

To further verify the effect of umbelliferone on *A. hydrophila* in vivo, we investigated the protective effect of umbelliferone on grass carp infected by *A. hydrophila*. After infection with *A. hydrophila*, all animals in the model group showed obvious clinical symptoms, including inappetence, lethargy, reduced general activity, unresponsiveness, and staying at the bottom of the tank. Compared with the infection model group (A.H), grass carps in the low-dose group (A.H + UM 50 mg/kg) had a roughly 50% increase in appetite and improved mental state. The fish in the blank control group and the high-dose group (A.H + UM 100 mg/kg) had good mental states, sensitive reactions, and normal appetite. At 48 h postinfection of *A. hydrophila*, grass carps in the infection model group (A.H) showed redness, swelling, and bleeding at the injection site of the pectoral fin, abdominal enlargement, and even severe ascites. Both doses of umbelliferone significantly alleviated the bleeding at the injection site of the pectoral fin and ascites caused by *A. hydrophila* (Figure 5A).

After an intraperitoneal injection of *A. hydrophila*, the survival of grass carps in each group was recorded every 12 h until 48 h. The results are shown in Figure 5B. Survival in the infection model group (A.H) rapidly decreased after 12 h, and the final survival at 48 h was only 20.80%. The preventive administration of umbelliferone can significantly increase the survival of grass carps. The 48 h survivals were 61.38% in the high-concentration group (A.H + UM 100 mg/kg) and 44.80% in the low-concentration group (A.H + UM 50 mg/kg).

Forty-eight hours after challenge with *A. hydrophila*, the heart, hepatopancreas, spleens, mesonephros, and head kidneys were harvested for enumeration of bacteria. The results (Figure 5C–G) showed that umbelliferone could significantly reduce the bacterial load in all these tissues.

Histopathological examination showed that *A. hydrophila* could cause noticeable pathological changes in the heart, spleen, and hepatopancreas of grass carps. In the infection model group, the cytoplasm of dispersed cardiomyocytes was coagulated (Figure 6A). The endothelial cells swelled, and the myocardial fibers were broken. Meanwhile, abundant hemosiderin granules were deposited in the spleen (Figure 6B). In addition, as shown in Figure 6C, apparent congestion of hepatic sinusoids was observed. The hepatocytes around the blood vessels were swollen and structurally blurred. Cytoplasmic pyknosis and structural disintegration were visible in pancreatic acinar cells (Figure 6D). Administration of umbelliferone obviously improved these pathological damages caused by *A. hydrophila*.

## 3. Discussion

In this study, we researched the effects of umbelliferone on various virulence of *A. hydrophila* at sub-MICs. Biofilm formation is a crucial virulence property. The protective effect of biofilm on bacteria can make the bacteria in the biofilm withstand the damage of pH change, nutrient deficiency, and mechanical force in the surrounding environment. It is also considered an essential factor for the occurrence of multidrug resistance of bacteria [15]. In this study, the biofilm formation test indicated that umbelliferone significantly inhibited the formation of *A. hydrophila* biofilm, with the highest inhibition rate of 53.8% at 48 h. Similar antibiofilm activity against other bacteria was also observed [16]. It is well established that multiple factors can influence bacterial biofilm formation, one of which is bacterial motilities. Guided by hydrodynamics, bacteria can be reversibly attached to ideal surfaces by swimming in a low agar environment of 0.3–0.4%. Swarming occurs commonly on the surface of more viscous semisolid media (0.5–0.7% agar), driven by multiple flagella and requiring specific bacterial population density and nutrients [17,18]. Monte’s [19] research showed that umbelliferone reduced *E. coli* swimming, swarming motilities, and biofilm formation. In this experiment, umbelliferone inhibited the two motilities of *A. hydrophila*, which was consistent with the results of biofilm formation. *A. hydrophila* can produce and secrete hemolysin, often causing hemolysis of mammalian and fish red blood cells. The capacity of *A. hydrophila*-induced hemolysis is mainly related to its secreted exotoxins, including enterotoxin, cytotoxin, and aerolysin [20]. Our results showed that the inhibition rate of umbelliferone on *A. hydrophila*-induced hemolysis was up to 95.25% at 128 μg/mL. The inhibitory effect of coumarins on *Vibrio brilliant*-induced hemolysis has also been reported [21].

A recent study has shown that coumarins are potential quorum-sensing inhibitors [22]. The quorum-sensing system is a bacterial cell-to-cell signaling system that regulates the production of virulence factors through signaling molecules. We found that umbelliferone reduced the production of violacein pigment by *C. violaceum* CV026 (AHL reporter strain) and the bioluminescence intensity of *V. Harvey* BB170 (AI-2 reporter strain), suggesting that umbelliferone could inhibit AHL and AI-2 production in *A. hydrophila*. According to these results, we speculated that the quorum-sensing inhibitory effect of umbelliferone on *A. hydrophila* might be one of the ways to inhibit bacterial virulence.

To further investigate the antivirulence mechanism of umbelliferone on *A. hydrophila*, the gene expression profiles in *A. hydrophila* were compared before and after treatment with umbelliferone. We found that some genes related to bacterial exotoxin, the two-component system, and the secretion system are downregulated post umbelliferone treatment. Among these differently differentially expressed genes, virulence-related genes are of the most concern. On the one hand, umbelliferone directly downregulated exotoxin genes encoding enterotoxin, RTX toxin, and heat-stable hemolysin, which play a vital role in the pathogenesis of *A. hydrophila* [23]. On the other hand, umbelliferone downregulated two systems (the two-component system and the secretion system) involved in controlling bacterial virulence factors. The two-component system in bacteria can regulate the expression of many virulence-related genes to help bacteria respond to environmental stimuli [24] and evade the host immune system [25]. The secretion system is the cellular device used by pathogenic bacteria to secrete virulence factors to invade the host cells. It has received a great deal of attention from microbiologists in recent years. In our experiment, the expression levels of some genes coding the type II secretion system (T2SS) and the type VI secretion system (T6SS) were downregulated after treatment with umbelliferone. T2SS is mainly involved in bacteria adhesion and the invasion of the host cells, secreting protease and cytotoxic enterotoxin, ultimately leading to host cell damage [26,27]. *ExeI* and *exeN* encode proteins containing procollagen signaling peptides in the type II secretion system. Umbelliferone can reduce the expression of both genes in *A. hydrophila*. T6SS can directly deliver effectors to target cells, participating in interbacterial competition and pathogenesis [28]. Our results indicated that the T6SS-related genes (e.g., *PAAR*, *vgrG*, *vasH*, *tssK*, and *tssM*) were downregulated in the umbelliferone treatment group. VgrG is expected to sit at the top of the Hcp tube and propel toward the target cell. The PAAR-repeat protein binds to the end of the VgrG trimer, forming the conical end of the Hcp-VgrG puncture structure [29]. TssK is a baseplate component, and TssM is a membrane complex [30,31]. Combining the above literature with our results, we speculate that the T6SS system is one of the critical targets for umbelliferone to reduce toxin secretion from *A. hydrophila*. Iron regulates the metabolic process and the pathogenicity of bacteria, which is closely related to biofilm formation and the pathogenicity of bacteria [32,33]. ExbB-ExbD is a protein complex that harvests energy and transmits it via TonB to TonB-dependent outer membrane receptors to trigger iron uptake [34]. After treatment with umbelliferone, the expression of genes, including *TonBs*, *ExbB*, and *ExbD*, were downregulated, indicating that the drug can reduce iron uptake in *A. hydrophila*.

The results of the in vivo experiment further confirmed the protective effect of umbelliferone on infected animals with *A. hydrophila*. Early administration of umbelliferone could significantly reduce the mortality of grass carps infected by *A. hydrophila*, decrease the bacterial load in tissues, and effectively alleviate symptoms. These results demonstrated that prophylactic administration of umbelliferone could be helpful in eliminating bacteria and attenuating *A. hydrophila* infection. However, umbelliferone can only partially eliminate bacteria in tissues, suggesting that there are other important ways to resist infection with *A. hydrophila* in vivo, such as decreasing the virulence to attenuate bacterial pathogenicity.

## 4. Materials and Methods

### 4.1. Bacterial Strains, Chemical Reagents, and Cultivation

The clinical isolate *A. hydrophila* was kindly provided by Prof. Defang Chen (Sichuan Agriculture University) and was used throughout the in vitro and in vivo study. Umbelliferone (purity >98%) was purchased from Shanghai Yuan Ye Biotechnology Co., Ltd. Umbelliferone was dissolved in 100% DMSO to make 80 mg/mL and stored at −20 °C. The stock solution was diluted with culture medium or PBS before use. All bacterial culture mediums were purchased from Haibo Bio (Qingdao, China). *A. hydrophila* were cultured in TSB or on TSA at 30 °C unless otherwise stated.

### 4.2. Determinations of Minimum Inhibitory Concentration (MIC) and Growth Curve

The MIC was determined by the microdilution broth method. To measure the growth curve of *A. hydrophila*, the bacterial suspension was cultured with continuous shaking in TSB at 30 °C with or without umbelliferone, and the optical density at 600 nm of the mixture was measured every 2 h.

### 4.3. Biofilm Formation

The crystal violet staining method was used to determine biofilm formation. Overnight-cultured bacteria were diluted at 1:100 in TSB, and 200 µL was then added to a 96-well plate and incubated with various concentrations of umbelliferone at 30 °C. Cultures were sampled after 12 h, 24 h, 48 h, and 72 h, and the OD600 values of the cultures were determined. Nonadhered bacteria were removed and gently rinsed three times with PBS. The biofilm was fixed in methanol for 15 min and stained with 1% crystal violet for 30 min. The excess stain was washed away with distilled water and dried. Bound crystal violet was dissolved with 100 µL of 33% acetic acid for 30 min, and OD values of these solutions were measured at 590 nm. The ratio of OD590/OD600 was used to determine biofilm production.

### 4.4. Motility Assays

LB medium with 0.3% agar (*wt*/*vol*) was used for the swimming motility assay, and LB medium with 0.5% agar (*wt*/*vol*) for the swarming motility assay. The sub-MICs of umbelliferone were added to the swimming medium or swarming medium. After drying, 5 μL of overnight bacterial suspensions were stabbed into the plates and incubated at 30 °C for designated times. The diameter of the migration zone of bacteria was measured.

### 4.5. Hemolysis Assay

The hemolysis assay was performed using the modified method of Jing Dong et al. [35]. Overnight-cultured bacteria were diluted at 1:100 in TSB, then sub-MICs of umbelliferone were added to the bacterial suspension and incubated at 30 °C with shaking at 160 r/min for 17 h. Simultaneously, the blank control group (only with medium), the bacterial group (without drugs), and the solvent control group were set up. The cocultures were centrifuged at 4500 r/min for 10 min. Next, 60 μL of supernatant was added to 940 μL of 5% rabbit erythrocyte suspension and incubated in an incubator at 37 °C for 20 min. The supernatant was obtained by 10,000 r/min centrifugation for 2 min, and the absorbance was determined at 543 nm.

### 4.6. Determination of Quorum-Sensing Signals

The effect of umbelliferone on AHL quorum-sensing signals was detected using *C. violaceum* CV026 as an indicator. *A. hydrophila* treated with or without umbelliferone under sub-MICs were incubated for 12 h at 30 °C with shaking, then were centrifuged at 7000 r/min for 5 min to obtain the supernatant. The supernatant was extracted by an equal volume of ethyl acetate (containing 0.5% methanoic acid) three times. The upper organic phase was collected and dried in a vacuum drying oven. The dried residue was dissolved into 200 μL dimethyl sulfoxide and stored at 4 °C. The overnight cultures of *C. violaceum* CV026 were mixed at 1% (*vol*/*vol*) with LB solid medium (40 °C). Then, 30 μL of the extracted AHL was added into the hole on the agar plate and cultured overnight at 30 °C to observe whether there was a purple circle and measure the circle diameter.

The effect of umbelliferone on AI-2 quorum-sensing signals was measured by using *V. harveyi* BB170 as an indicator. *A. hydrophila* treated with or without umbelliferone under sub-MICs were incubated for 12 h at 30 °C with shaking, then centrifuged at 4000 r/min for 10 min to obtain the supernatant. The supernatant was filtered with a 0.22 μm filter membrane and stored at 4 °C. *V. harveyi* BB170 was diluted by 1:1000 in fresh AB medium. Subsequently, 180 μL of the diluted *V. harveyi* BB170 solution and 20 μL of the supernatant of *A. hydrophila* cultures treated or untreated with umbelliferone were added to a white 96-well plate and incubated at 30 °C. After 0 h, 12 h, and 15 h, chemiluminescence was detected using a microplate reader (VarioskanFlash LUX, Thermo Fisher).

### 4.7. RNA Isolation and Sequencing

Total RNA was extracted from *A. hydrophila* cultures grown for 12 h with or without umbelliferone (128 μg/mL) treatment using TRIzol reagent (Thermo Fisher Scientific, Carlsbad, CA) following the manufacturer’s instructions. RNA integrity was assessed using Bioanalyzer 2100 system (Agilent Technologies, CA, USA), followed by mRNA being purified from total RNA using probes to remove rRNA. Following fragmentation, the cDNA library was constructed and sequenced on an Illumina Novaseq platform at the Novogene Bioinformatics Institute (Beijing, China).

### 4.8. Quantification of Gene Expression Level

Clean reads were obtained by removing reads containing adapter, reads containing ploy-N, and low-quality reads from raw data and were then mapped to the *A. hydrophila* genome (RefSeq assembly accession: GCF_000963645.1) using bowtie2 (ver. 2.3.4.3) [36]. HTSeq (ver. 0.9.1) was used to count the read numbers mapped to each gene [37], and the fragments per kilobase of exon model per million mapped reads (FPKM) for each gene were then calculated based on the length of the gene and the read counts mapped to the gene.

### 4.9. Differential Expression and Functional Enrichment Analyses

Differential expression analysis was performed using R package DESeq2 (ver. 1.20) [38]. Genes with an adjusted *p*-value of <0.05 found by DESeq were assigned as differentially expressed. Gene ontology (GO) enrichment analysis and Kyoto Encyclopedia of Genes and Genomes (KEGG) pathway analysis of differentially expressed genes were implemented by the R package cluster profile (ver. 3.8.1) [39].

### 4.10. qRT-PCR

The reliability of transcriptome data was validated by qRT-PCR of 12 selected genes. Total RNA was harvested from control and sub-MICs of umbelliferone-treated cultures of *A. hydrophila* strains using TRIzol reagent (Thermo Fisher Scientific, Carlsbad, CA, USA), then RNA was reversed transcribed to cDNA by PrimeScriptTM RT Reagent Kit with gDNA Eraser (Perfect Real Time). Real-time PCR was conducted using TB Green Premix EX TaqTM (Tli RNaseH Plus) according to the manufacturer’s protocol. The normalized gene expression levels were calculated with the 2^−ΔΔCT^ method using the *apoA* gene as the endogenous control. The primer pairs used in this study are shown in Appendix A.

### 4.11. Animal Experiments

All animal experiments were conducted under the guidance of the Animal Welfare and Ethics Committee of Sichuan Agricultural University (Permit No. DKY-S20176908). Grass carps (body length: 10–15 cm, weight: 23–27 g) were obtained from Qianhu Aquatic Fish Fry Mall. A total of 120 grass carps were acclimated for 7 days and fed with a commercial feed (Tongwei, China) (composition: 30.0% protein, 4.0% lipid, 12.0% carbohydrate, and 15.0% ash) three times a day at 3% of body weight. Water was partly replaced daily, and the water temperature was maintained at 21 ± 1 °C with a heating rod, continuously aerated with an air pump. All fish were maintained on a 12:12 hour light: dark cycle. Grass carps were randomly divided into four groups, including two umbelliferone treatment groups (50 and 100 mg/kg·bw), the infection model group, and the blank control group. Three replicates (each with 10 fish) were performed for each group. Dexamethasone was injected intraperitoneally with 200 μg per fish for 3 days. Seventy hours later, drug treatment groups, the infection model group, and the blank control group were intraperitoneally administrated with 0.1 mL of umbelliferone, DMSO, and PBS, respectively. Two hours post above treatment, *A. hydrophila* was intraperitoneally challenged with 5 × 10^8^ CFU/mL in the infection model and drug treatment groups. The clinical symptoms and the mortalities were recorded at 12 h, 24 h, 36 h, and 48 h after bacterial challenge. In addition, bacterial load assessment and histopathological examination were respectively performed by random selection of three grass carps from each replicate at 48 h postchallenge.

Assessment of bacterial loads in tissues. Under aseptic conditions, the heart, hepatopancreas, spleen, mesonephros, and cephalic kidneys were homogenized and diluted in PBS. 0.1 mL diluent was coated on *A. hydrophila* isolation medium (RS solid medium) and incubated at 30 °C for 24 h. Colonies were counted to calculate the average number of CFU/g.

Histopathological examination. Hearts, spleens, and hepatopancreases were removed, washed with normal saline, and fixed in a 4% paraformaldehyde solution. After gradient alcohol dehydration, the tissues were embedded in paraffin wax. Histological sections (5 μm) were prepared and stained with hematoxylin and eosin. Zeiss Axio Imager system (Zeiss, Oberkochen, Germany) was used for microscopic imaging of tissue sections.

### 4.12. Statistical Analysis

Statistical analysis was performed using IBM SPSS 23.0 software (IBM Corp.). Data were expressed as mean ± SEM, and comparisons between the two groups were performed using Student’s *t*-test. *p* < 0.05 was considered statistically significant.

## 5. Conclusions

The sub-MICs of umbelliferone could significantly inhibit the virulence of *A. hydrophila* in vitro, increase the survival rate of grass carps infected with *A. hydrophila*, and effectively reduce the bacterial load in the viscera. The two-component system and secretory system may be the potential pathways of umbelliferone to reduce the virulence of *A. hydrophila*. Umbelliferone may be a promising drug candidate for the prevention/treatment of *A. hydrophila* infection in aquatic animals.

## Figures and Tables

**Figure 1 ijms-23-11119-f001:**
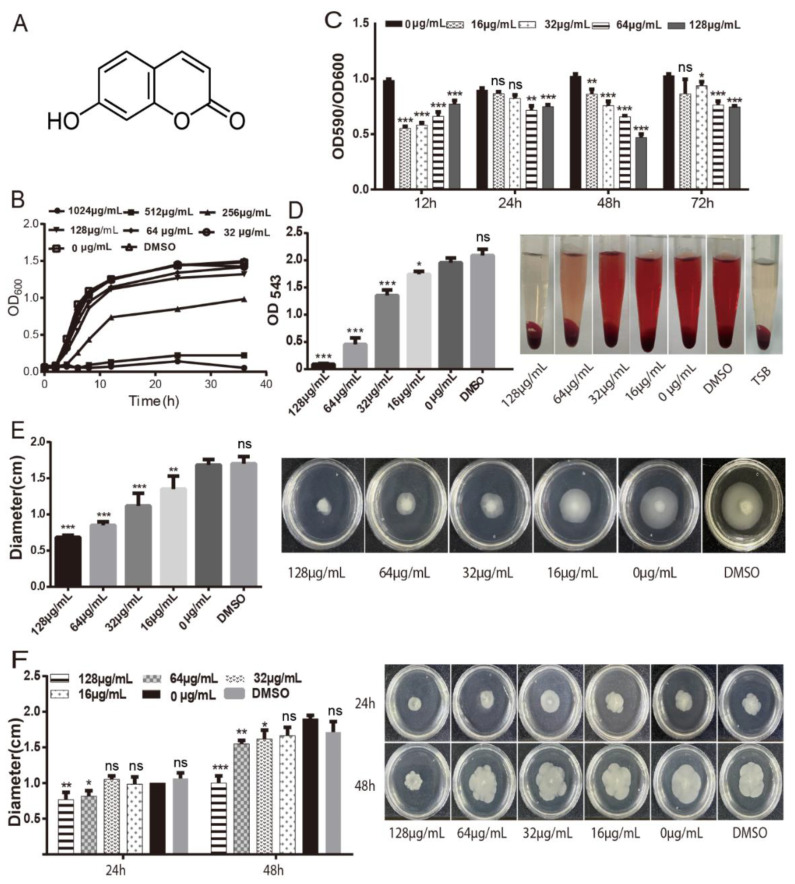
Effect of umbelliferone on the virulence of *A. hydrophila*: (**A**) structure of umbelliferone; (**B**) growth curve of *A. hydrophila*; (**C**) biofilm formation of *A. hydrophila* was detected by the crystal violet staining method; (**D**) effect of umbelliferone on hemolysis activity of *A. hydrophila*; (**E**) effect of umbelliferone on swimming motility of *A. hydrophila*; (**F**) effect of umbelliferone on swarming motility of *A. hydrophila*; ***, *p* < 0.001; **, *p* < 0.01, *, *p* < 0.05; ns, not significant, compared with the control (0 μg/mL).

**Figure 2 ijms-23-11119-f002:**
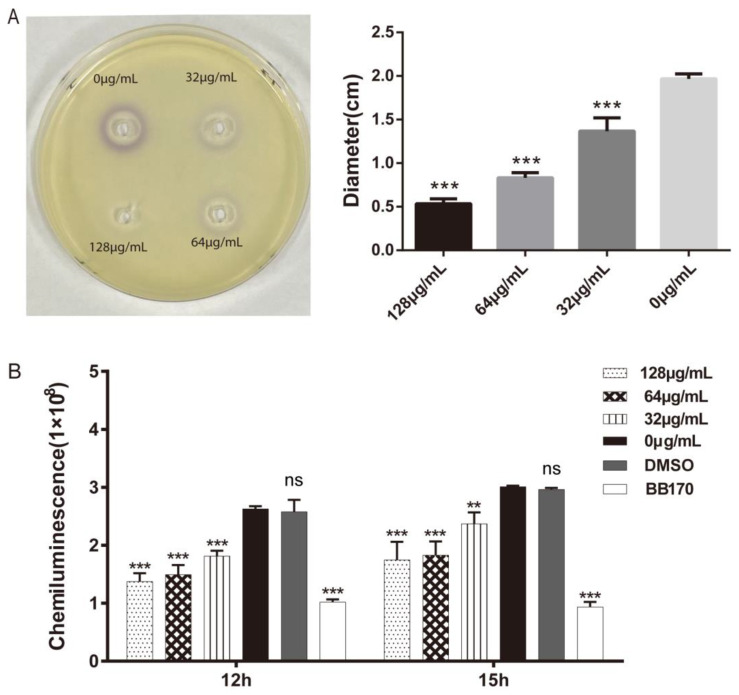
The effect of umbelliferone on the quorum-sensing signals of *A. hydrophila*. (**A**) Umbelliferone inhibited AHL production in *A. hydrophila*, which reduced violacein synthesis in *C. violaceum* CV026. (**B**) Umbelliferone inhibited AI-2 production in *A. hydrophila*, which reduced bioluminescence intensity in *V. harveyi* BB170. ***, *p* < 0.001; **, *p* < 0.01; ns, not significant, compared with the control (0 μg/mL).

**Figure 3 ijms-23-11119-f003:**
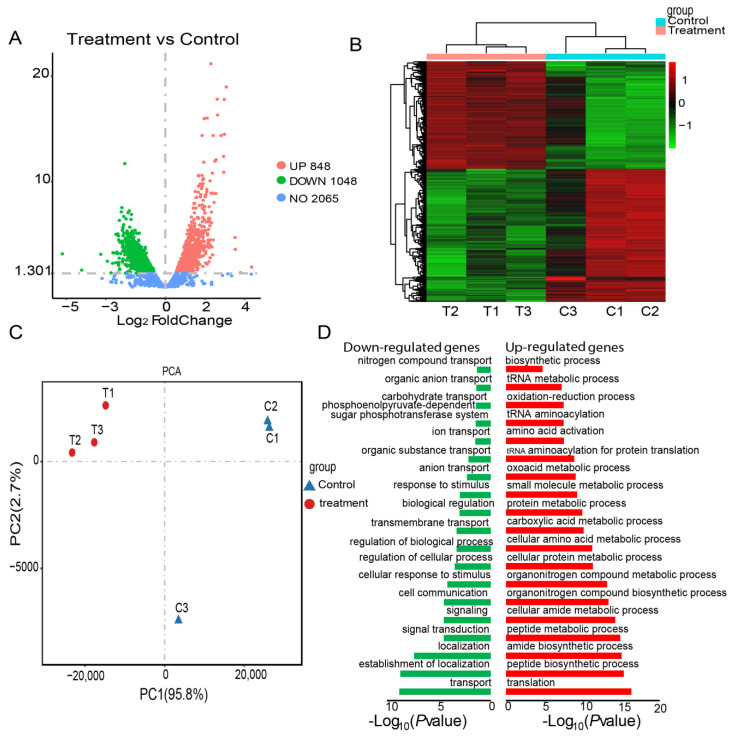
Transcriptome changes in *A. hydrophila* after umbelliferone treatment: (**A**) volcano plot of differentially expressed genes; (**B**) heatmap of the differentially expressed genes in *A. hydrophila* treated or untreated with umbelliferone; (**C**) PCA plot of differentially expressed genes. Control and umbelliferone treatment samples are shown in cyan and red, respectively; (**D**) functional enrichment analysis plot. Representative GO terms of up- and downregulated differentially expressed genes.

**Figure 4 ijms-23-11119-f004:**
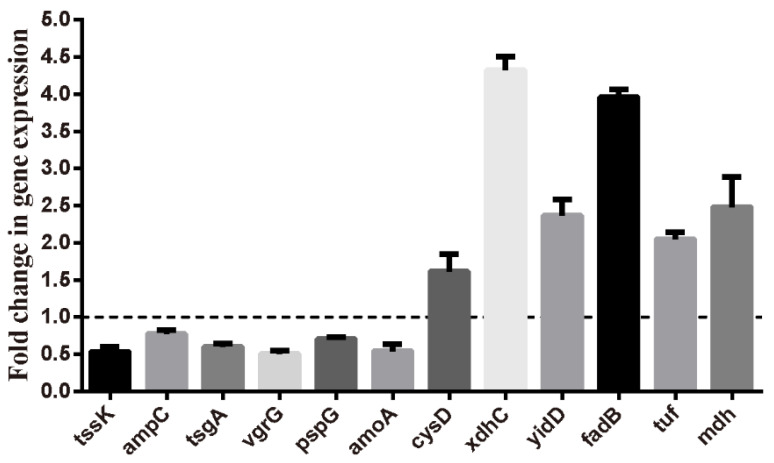
Validation of expression levels of 12 randomly selected genes by qRT-PCR compared with the transcriptome results. Results are represented by mean ± SEM (*n* = 3).

**Figure 5 ijms-23-11119-f005:**
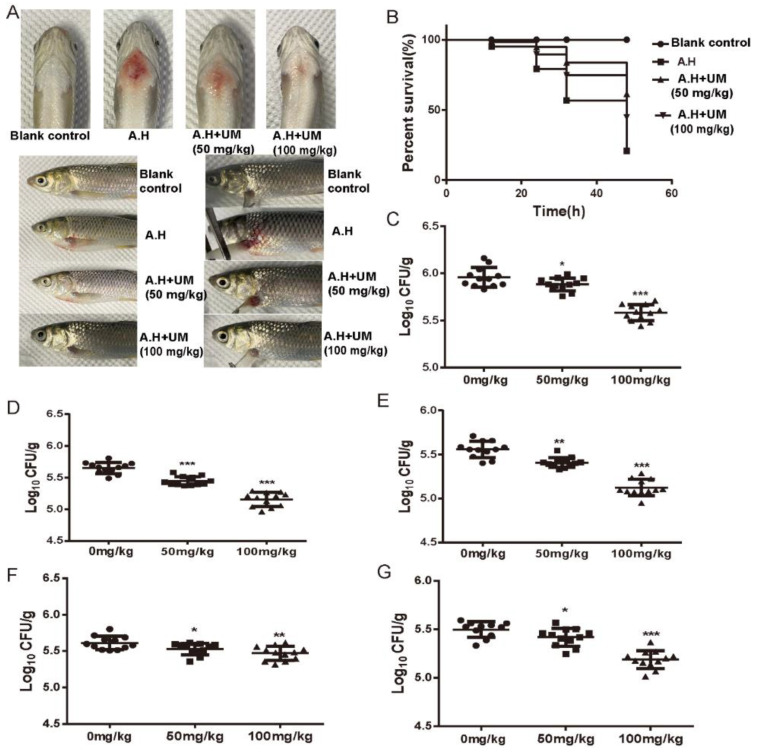
Protective effect of umbelliferone on grass carp infected with *A. hydrophila*. (**A**) Umbelliferone alleviates the symptoms of grass carps infected by *A. hydrophila*. (**B**) Umbelliferone increases the survival rate of grass carps infected with *A. hydrophila*. The bacterial load in (**C**) heart, (**D**) hepatopancreas, (**E**) spleen, (**F**) Mesonephros, and (**G**) Head kidney of grass carps with umbelliferone treatment decreased significantly. ***, *p* < 0.001; **, *p* < 0.01, *, *p* < 0.05, compared with the infection model group (0 mg/kg).

**Figure 6 ijms-23-11119-f006:**
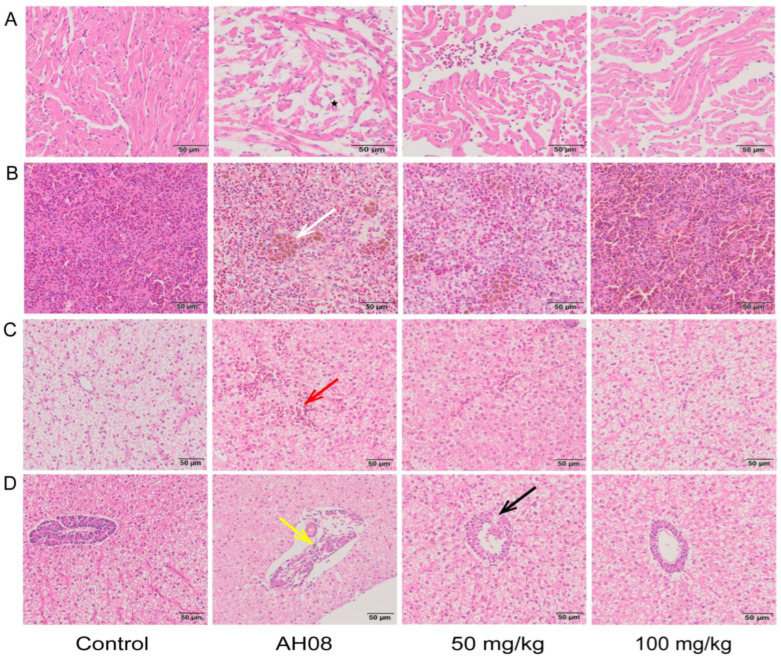
Histopathological analysis for selected tissues from various groups. Umbelliferone alleviated *A. hydrophila*-induced heart (**A**), spleen (**B**), and hepatopancreas (**C**,**D**) injury in grass carps. The sections were stained with H&E. Cardiomyocytes were dispersed (black star), deposited hemosiderin granules in the spleen (white arrows), apparent congestion of hepatic sinusoids (red arrow), cytoplasmic pyknosis (yellow arrow), and mild pericardial edema (black arrow).

**Table 1 ijms-23-11119-t001:** The representative downregulated gene list.

Gene Name	Log_2_ Fold Change	Description
Genes associated with exotoxin		
*VU14_RS08845*	−1.6900	HBL/NHE enterotoxin family protein
*VU14_RS08840*	−2.304	Bacillus hemolytic enterotoxin (HBL)
*rtxA*	−1.5406	MARTX multifunctional-autoprocessing repeats-in-toxin holotoxin RtxA
*VU14_RS05570*	−1.5204	Thermostable hemolysin
Genes associated with two-component system		
*VU14_RS20395*	−1.8170	Two-component system response regulator
*VU14_RS05225*	−1.5966	Response regulator receiver domain
*yehT*	−1.6300	Two-component system response regulator BtsR
*uhpB*	−1.5975	Signal transduction histidine-protein kinase/phosphatase UhpB
Genes associated with secretory systems		
*exeI*	−1.6425	GspI family T2SS minor pseudopilin variant ExeI
*exeN*	−1.5519	GspN family type II secretion system protein ExeN
*VU14_RS12995*	−2.4463	Type VI secretion system ImpA family
*VU14_RS12990*	−2.0408	Type VI secretion system PAAR protein
*vgrG*	−1.7650	Type VI secretion system effector VgrG
*tssK*	−1.7623	Type VI secretion system baseplate subunit TssK
*tssM*	−1.6745	Type VI secretion system membrane subunit TssM
*vasH*	−1.1018	σ-54-dependent transcriptional regulator VasH
*VU14_RS13025*	−1.5086	Type VI secretion system protein DotU
Genes associated with iron transport		
*VU14_RS01835*	−2.1770	Ligand-gated channel protein
*VU14_RS04515*	−1.8007	Energy transducer TonB
*VU14_RS20175*	−1.7896	TonB-dependent siderophore receptor
*VU14_RS04520*	−1.8444	Biopolymer transporter ExbD
*VU14_RS14250*	−1.7692	TonB-dependent hemoglobin/transferrin/lactoferrin family receptor
*VU14_RS17650*	−1.7498	TonB-dependent hemoglobin/transferrin/lactoferrin family receptor
*VU14_RS20010*	−1.5912	TonB-dependent copper receptor
*VU14_RS12465*	−1.5400	TonB-dependent siderophore receptor
*amoA*	−1.5178	Isochorismate synthases AmoA
*amoG*	−1.5635	Amonabactin biosynthesis nonribosomal peptide synthetase AmoG
*amoH*	−1.5880	Amonabactin biosynthesis glycine adenylation protein AmoH

**Table 2 ijms-23-11119-t002:** The representative upregulated gene list.

Gene Name	Log_2_ Fold Change	Description
Genes associated with ribosomes		
*rpsG*	2.2465	30S ribosomal protein S7
*rpsF*	2.2355	30S ribosomal protein S6
*rpsJ*	2.1849	30S ribosomal protein S10
*rpsR*	2.1632	30S ribosomal protein S18
Genes associated with citrate cycle (TCA cycle)		
*mdh*	2.3949	Malate dehydrogenase
*VU14_RS12215*	2.6071	Fumarate hydratase
*frdD*	2.3141	Fumarate reductase subunit FrdD
*sucD*	1.9545	Succinate-CoA ligase subunit α
Genes associated with carbon metabolism		
*VU14_RS04850*	2.2821	Glyceraldehyde-3-phosphate dehydrogenase
*fadB*	2.4898	Fatty acid oxidation complex subunit α FadB
*VU14_RS06340*	1.6773	Formate dehydrogenase subunit α
Genes associated with oxidative phosphorylation		
*frdD*	2.3141	Fumarate reductase subunit FrdD
*sdhD*	1.7136	Succinate dehydrogenase hydrophobic membrane anchor protein
*sdhC*	1.5912	Succinate dehydrogenase cytochrome b556 subunit

## Data Availability

The RNA-seq datasets are available in the Genome Sequence Archive (Genomics, Proteomics & Bioinformatics 2021) in the National Genomics Data Center (Nucleic Acids Res 2022), China National Center for Bioinformation/Beijing Institute of Genomics, Chinese Academy of Sciences (GSA: CRA007591).

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
