# Peer review of "The Antivirulence Activity of Umbelliferone and Its Protective Effect against A. hydrophila-Infected Grass Carp"

_ijms, 2022, doi:10.3390/ijms231911119_

Round 1
Reviewer 1 Report
The effect of umbelliferone on the pathogenicity of A. hydrophila in vitro and in vivo in grass carp infected with the bacterium was studied. The effect of umbelliferone on the inhibition of A. hydrophila virulence, such as hemolysis, biofilm formation, swimming and swarming motility, and its AHL and AI-2 quorum sensing signals, was evaluated. Its mechanism of action was analyzed through information from the transcriptome. Umbelliferone is suggested as a potential drug for the treatment of A. hydrophila infection. There are quite a few reports of similar studies done on other bacteria, though, it is clearly addressed and the results are conclusive. However, I think it is important to develop the idea of the probability that umbelliferone could actually be used as a drug to treat bacterial infections, to give a sense of novelty to this type of study.
Author Response
Dear editor and reviewers,
We would like to thank the honorable editor and reviewers for considerate attention and constructive suggestions and comments on our manuscript entitled “The anti-virulence activity of umbelliferone and its protective effect against A. hydrophila infected grass carp”.
We have gone through all of the editors’ and reviewers’ comments in detail and done our best to address all of their questions and suggestions. Below we provide our point-to-point responses, and hope that they are satisfactory.
We look forward to hearing a positive response from you.
Best regards,
Dr. Xun Wang
Sichuan Agricultural University, China
Reviewer 1
Comment:
The effect of umbelliferone on the pathogenicity of A. hydrophila in vitro and in vivo in grass carp infected with the bacterium was studied. The effect of umbelliferone on the inhibition of A. hydrophila virulence, such as hemolysis, biofilm formation, swimming and swarming motility, and its AHL and AI-2 quorum sensing signals, was evaluated. Its mechanism of action was analyzed through information from the transcriptome. Umbelliferone is suggested as a potential drug for the treatment of A. hydrophila infection. There are quite a few reports of similar studies done on other bacteria, though, it is clearly addressed and the results are conclusive. However, I think it is important to develop the idea of the probability that umbelliferone could actually be used as a drug to treat bacterial infections, to give a sense of novelty to this type of study.
Response:
Thank you very much for your kind comments.
Reviewer 2 Report
The title of the article is relevant and informative. The research question and motivation are clearly justified with the relevant problem statement. The study method is valid and reliable with well-defined variables. The findings obtained were presented in an organized way and discussed from multiple angles and placed into the context without being over interpreted.
However, there are queries that authors must look into to improve the quality of the manuscript. The paper can be accepted after major revision.
o Provide a brief description umbelliferone in introduction, why it is important, from which group it belongs etc.
o Provide a brief hypothesis of the study in introduction
o Line 68-72: Restructure and add more information about the study
o Figures: label the control properly both in figure and legend. Now it’s difficult to say whether control is DMSO or BB120
o Figure 6: add arrow mark with different colour to highlights the changes in histological section. Write only control instead of blank control
o Again in materials and methods: it’s better to provide a paragraph about Umbelliferone. From which group it belongs, how the stock culture was prepared, what is the possible efficacy, etc.
o Line 375: 120 numbers of grass carp
o Add composition of feed, whether prepared by self or commercial?
o It’s very difficult to understand Animal experiments. It’s better to rewrite it.
o It’s better to check the whole manuscript thoroughly for possible grammar mistake and syntax error before resubmission of revised manuscript.
Author Response
Dear editor and reviewers,
We would like to thank the honorable editor and reviewers for considerate attention and constructive suggestions and comments on our manuscript entitled “The anti-virulence activity of umbelliferone and its protective effect against A. hydrophila infected grass carp”.
We have gone through all of the editors’ and reviewers’ comments in detail and done our best to address all of their questions and suggestions. Below we provide our point-to-point responses, and hope that they are satisfactory.
We look forward to hearing a positive response from you.
Best regards,
Dr. Xun Wang
Sichuan Agricultural University, China
Detailed responses to reviewers
All comments made by the editor and reviewers are in gray italics, and our responses are in black. Revisions are marked in red color in response letter.
Comments from the reviewers:
Reviewer 2
Comment 2-1:
The title of the article is relevant and informative. The research question and motivation are clearly justified with the relevant problem statement. The study method is valid and reliable with well-defined variables. The findings obtained were presented in an organized way and discussed from multiple angles and placed into the context without being over interpreted.
However, there are queries that authors must look into to improve the quality of the manuscript. The paper can be accepted after major revision.
Response 2-1:
We appreciate for reviewer 2’s comments.
Comment 2-2:
Provide a brief description umbelliferone in introduction, why it is important, from which group it belongs etc..
Response 2-2:
Thank you for your suggestion. We’ve complemented the content you point out.
Lines 68-73:
Umbelliferone(7-hydroxycoumarin) is a kind of coumarin derivatives with a hydroxyl substitute at position C7 widely derived from plants. Among various coumarin derivatives, hydroxycoumarin is considered to be a type with relatively prominent antimicrobial activity. Yang Liang et al. tested the inhibitory effects of 18 kinds of coumarins against the plant pathogen R. solanacearum, and found that umbelliferone had better antibacterial activity than coumarin.
Comment 2-3:
Provide a brief hypothesis of the study in introduction.
Response 2-3:
Done as requested.
Lines 73-77:
Although there have been some studies on the activity of umbelliferone against bacteria such as R. solanacearum and E. coli, its effect on aquatic animal pathogens has not been reported. In our preliminary study, umbelliferone exhibited a significant inhibition effect on the quorum sensing system of A. hydrophila, suggesting that umbelliferone has potential anti-virulence activity.
Comment 2-4:
Line 68-72: Restructure and add more information about the study.
Response 2-4:
Done as requested.
Lines 77-82:
Here, we explored the effect of umbelliferone on the virulence factors of A. hydrophila , and analyzed its potential mechanism through transcriptome information. Furthermore, the protective effect of umbelliferone on grass carp infected with A. hydrophila was studied. Our results demonstrated that umbelliferone had an obvious inhibitory effect on the pathogenicity of A. hydrophila in vitro and in vivo.
Comment 2-5:
Figures: label the control properly both in figure and legend. Now it’s difficult to say whether control is DMSO or BB120.
Response 2-5:
Thank you for pointing out this. According to your suggestion, we’ve clearly labeled the control group in figure and legend to avoid confusion.
Comment 2-6:
Figure 6: add arrow mark with different colour to highlights the changes in histological section. Write only control instead of blank control.
Response 2-6:
Done as requested.
Comment 2-7:
Again in materials and methods: it’s better to provide a paragraph about Umbelliferone. From which group it belongs, how the stock culture was prepared, what is the possible efficacy, etc..
Response 2-7:
According to your suggestion, we’ve complemented preparation method of the stock culture.
Lines 290-292:
Umbelliferone were dissolved in 100% DMSO to make 80 mg/mL and stored at -20℃. The stock solution was diluted with culture medium or PBS before use.
Comment 2-8:
Line 375: 120 numbers of grass carp.
Response 2-8:
Done as requested.
Comment 2-9:
Add composition of feed, whether prepared by self or commercial?
Response 2-9:
We’ve added the composition of feed.
Lines 382-384:
120 numbers of grass carps were acclimated for 7 days, and fed with a commercial feed (Tongwei, China) (composition: 30.0% protein; 4.0% lipid; 12.0% carbohydrate and 15.0% ash) three times a day at 3% of body weight.
Comment 2-10:
It’s very difficult to understand Animal experiments. It’s better to rewrite it..
Response 2-10:
We’ve rewrote animal experiments section according to your suggestion.
Lines 387-397:
Grass carps were randomly divided into four groups, including two umbelliferone treatment groups (50 and 100 mg/kg·bw), the infection model group and the blank control group. 3 replicates (each with 10 fish) were performed for each group. Dexamethasone was injected intraperitoneally with 200 μg per fish 3 days. Seventy hours later, drug treatment groups, the infection model group and the blank control group were intraperitoneally administrated with 0.1 mL of umbelliferone, DMSO, and PBS, respectively. Two hours post above treatment, A. hydrophila was intraperitoneally challenged with 5×108 CFU/mL in infection model and drug treatment groups. The clinical symptoms and the mortalities were recorded at 12 h, 24 h, 36 h, and 48 h after bacterial challenge. In addition, bacterial load assessment and histopathological examination were respectively performed by random selection of 3 grass carps from each replicate at 48 h post-challenge.
Comment 2-11:
It’s better to check the whole manuscript thoroughly for possible grammar mistake and syntax error before resubmission of revised manuscript..
Response 2-11:
Done as requested.